

# The effect of tides on nearshore environmental DNA

Ryan P. Kelly, Ramón Gallego and Emily Jacobs-Palmer

School of Marine and Environmental Affairs, University of Washington, Seattle, WA,
United States of America

## ABSTRACT

We can recover genetic information from organisms of all kinds using environmental
sampling. In recent years, sequencing this environmental DNA (eDNA) has become
a tractable means of surveying many species using water, air, or soil samples. The
technique is beginning to become a core tool for ecologists, environmental scientists,
and biologists of many kinds, but the temporal resolution of eDNA sampling is often
unclear, limiting the ecological interpretations of the resulting datasets. Here, in a
temporally and spatially replicated field study using ca. 313 bp of eukaryotic COI
mtDNA as a marker, we find that nearshore organismal communities are largely
consistent across tides. Our findings suggest that nearshore eDNA from both benthic
and planktonic taxa tends to be endogenous to the site and water mass sampled,
rather than changing with each tidal cycle. However, where physiochemical water
mass characteristics change, we find that the relative contributions of a broad range
of organisms to eDNA communities shift in concert.

## INTRODUCTION

As environmental DNA (eDNA) becomes an increasingly important tool in ecological
research (*Sigsgaard et al., 2016*; *Deiner et al., 2017*), it is critical to understand how
techniques for eDNA collection and analysis perform under real-world conditions (*Port
et al., 2016*). In particular, we must characterize the spatial and temporal resolution of
amplicon-sequencing studies in order to confidently identify ecological patterns in the field
(*O'Donnell et al., 2017*); like any sampling technique, eDNA can reveal a phenomenon only
where the effects of that phenomenon are sufficiently large to be detected above background
variation (e.g., among replicates or time points).

Most efforts to quantify the behavior of eDNA in the field have taken the form of
quantitative PCR (qPCR) studies, in which the concentration of a particular template DNA
is measured over space or time. Notable recent examples include documenting degradation
of DNA over tens of meters in the flow of artificial streams (*Jerde et al., 2016*), caging trout
and measuring eDNA concentration at intervals downstream (*Jane et al., 2015*), estimating
eDNA production and degradation over time in a static environment (*Sassoubre et al.,
2016*), and estimating production and decay rates of eDNA from both caged and wild char
in a field setting (*Wilcox et al., 2016*), among others (e.g., *Thomsen et al., 2012*; *Deiner &
Altermatt, 2014*; *Tillotson et al., 2018*). For small planktonic organisms whose entire bodies

Corresponding author
Ryan P. Kelly, rpkelly@uw.edu

are likely present in the sample itself, the roles of transport and degradation appear less critical (*Medinger et al., 2010*). In sum, although the precise findings vary by setting and details of the molecular assay employed, even with highly sensitive qPCR, the distance from its source that eDNA can reliably be detected appears to be small, on the order of 10–1,000 m.

By contrast, less work has focused on the behavior of eDNA as reflected in ecological amplicon-sequencing studies. Port and colleagues (*2016*) showed that vertebrate eDNA communities can be distinguished at intervals of 60 m in nearshore marine waters, and *O'Donnell et al. (2017)* suggested that a similar spatial scale (<75 m) pertains to a broader nearshore metazoan assay. These were each single-time-point snapshots of animal species in dynamic environments, however, and especially in marine and aquatic environments in which spatial and temporal scales are linked by bulk transport of water, fine spatial resolution could be obliterated by water movement.

Nearshore marine habitats are among the most physically dynamic and biologically diverse on earth (*Helmuth et al., 2006*). The movement of water associated with tide is a fundamental property of these environments (*Babson, Kawase & MacCready, 2006*), dramatically shaping the life histories and ecology of organisms that live there. Environmental DNA surveys hold particular promise for better understanding thousands of species that may co-occur at a single nearshore marine location. However, use of this technique in the intertidal zone requires a practical knowledge of the effects of tide on the presence of eDNA sequences. Also, more generally, the intertidal environment provides rigorous testing grounds in which to discern the origins of genetic material detected in eDNA surveys.

Given recent work suggesting that eDNA signals are predominantly highly localized in space and time (*Thomsen & Willerslev, 2015*, and references therein)—although in some circumstances, eDNA may travel some distance (*Deiner & Altermatt, 2014*)—we asked whether marine eDNA community composition changes over tidal cycles at a given location. A scenario in which eDNA communities change in unpredictable ways with each new tide would suggest an exogeneous origin for that DNA, such that DNA arrives at a site with incoming tides, drawn from a pool of organisms existing elsewhere. By contrast, consistent eDNA communities over multiple tidal cycles would strongly suggest an endogenous origin and highly localized signal.

Here, we find that nearshore COI eDNA community composition is not strongly influenced by tide, and instead remains largely consistent within each geographic location across multiple successive tides. However, where shifts in the physical and chemical aqueous environment occur, the eDNA community appears to change accordingly; this result is consistent across both planktonic and benthic taxa. It therefore seems likely that changes in aqueous habitat characteristics—not tide itself—yield changes in eukaryotic eDNA communities.

## METHODS

### Field sampling

Our study design aimed to distinguish the effects of tide from site-level community differences and from sampling error. Consequently, we sampled each of three geographic
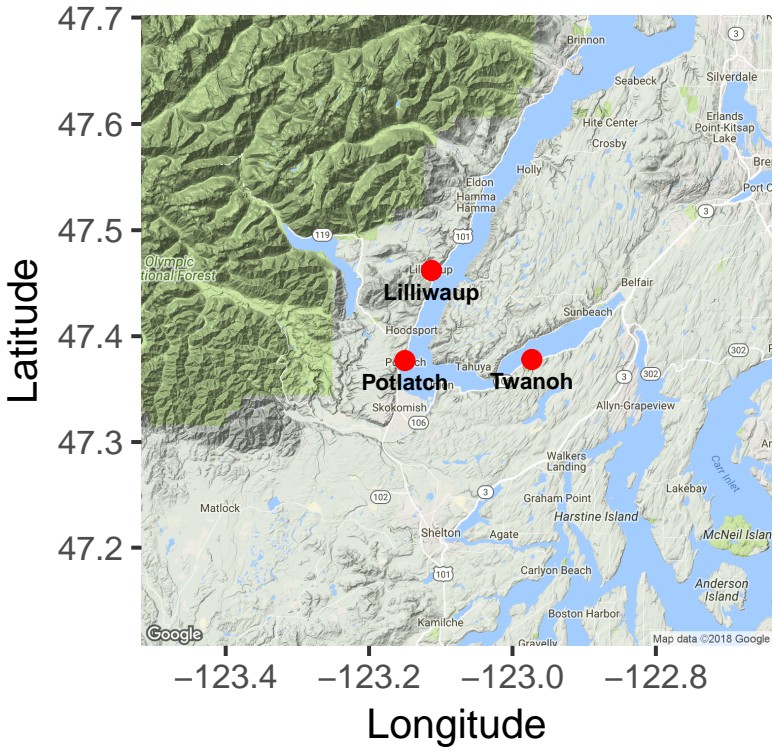

**Figure 1** Nearshore sampling locations in Hood Canal, Washington, USA.

locations (Fig. 1; GPS coordinates given in Table S1) in Hood Canal, Washington, USA, four times—twice during an incoming tide, and twice during an outgoing tide—over a ca. 28-hour period (Table 1). Despite its name, the Hood Canal is in fact a natural glacial fjord. We collected three 1-L water samples for eDNA analysis (ca. 10 m apart) at each site during each sampling event. No permits were required for collecting water samples, given the inherently public nature of saltwater in the United States. Each sample was collected at the surface (<1 m depth), using a ca. 3m-long pole with plastic collection bottle attached. We kept samples on ice until they could be processed, which occurred within hours of collection. We filtered 500 mL from each sample onto cellulose acetate filters (47 mm diameter; 0.45 um pore size) under vacuum pressure, and preserved the filter at room temperature in Longmire's buffer following *Renshaw et al. (2015)*. Deionized water served as a negative control for filtering. We measured water temperature and salinity with a hand-held multiprobe (model HI-9828; Hanna Instruments, Inc., Woonsocket, RI, USA), as well as measuring salinity with a handheld manual refractometer; the latter instrument more reliably reflected lab calibrations, and we use these measurements here.

## DNA extraction, amplification, and sequencing

We extracted total DNA from the filters using a phenol:chloroform:isoamyl alcohol protocol following *Renshaw et al. (2015)*, resuspended the eluate in 200 uL water, and used 1 uL of diluted DNA extract (1:10) as template for PCR. Although a single locus cannot completely characterize the biodiversity at a particular location (see, e.g., *Kelly et al., 2017*),

**Table 1 Samples by site and tide, showing balanced sampling design.** Each site ($N = 3$) had a total of four sampling events (time points), consisting of three water samples per event, and then 3–4 PCR replicates per water sample, such that we sequenced 36–44 individual PCR replicates per geographic sampling site. 35 of 36 samples were successfully processed, with 93 individual replicates survived quality-control, described below.

|           | Incoming tide | Outgoing tide |
| --------- | ------------- | ------------- |
| Lilliwaup | 5             | 6             |
| Potlatch  | 6             | 6             |
| Twanoh    | 6             | 6             |

we used a ca. 313 bp fragment of COI to assess the eukaryotic variance among our samples. This primer set (*Leray et al., 2013*) amplifies a broad array of taxa including representative diatoms, dinoflagellates, metazoans, fungi, and others; here, we simply use this primer set as an assay to characterize community similarity among samples. We followed a two-step PCR protocol to first amplify and then index our samples for sequencing, such that we could sequence many samples on the same sequencing run while avoiding amplification bias due to index sequence (*O'Donnell et al., 2016*). PCR mixes were 1X HotStar Buffer, 2.5 mM $MgCl_2$, 0.5 mM dNTP, 0.3 $\mu$M of each primer and include 0.5 units of HotStar Taq (Qiagen Corp., Valencia, CA, USA) per 20 $\mu$L reaction. The first round of PCR consisted of 40 cycles, including an annealing touchdown from 62 °C to 46 °C ($-1$ °C per cycle), followed by 25 cycles at 46 °C. The indexing PCR used a similar protocol with only 10 cycles at 46 °C.

We generated three PCR replicates for each of 35 water samples (three samples per sampling event, four sampling events per site, 3 sites = 36 water samples, of which 35 were processed successfully), and sequenced each replicate individually in order to assess the variance in detected eDNA communities due to stochasticity during amplification. We simultaneously sequenced positive (*Struthio camelus*—ostrich—tissue, selected because of the absence of this species in our study sites) controls with identical replication. We carried negative controls through amplification; no amplification was visible via gel electrophoresis in the negative controls, and fluorometry (Qubit; Thermo Scientific, Waltham, MA, USA) analysis showed negligible amounts of DNA present in those samples after amplification. We opted not to sequence the no-template controls for three reasons: first, there was no amplicon present in these samples, and carrying forward such samples is futile; second, one cannot generate equimolar libraries from samples with (i.e., experimental) and without (i.e., no-template control) amplicons, and therefore there is no straightforward way of comparing such samples quantitatively even if one did sequence no-template controls; and third, the purpose of no-template controls is to detect laboratory contamination and cross-contamination, and here, our positive controls and quality-control steps (see below) provided us a means of estimating and eliminating any such contamination prior to analysis.

Following library preparation according to manufacturers' protocols (KAPA Biosystems, Wilmington, MA, USA; NEXTflex DNA barcodes; BIOO Scientific, Austin, TX, USA), sequencing was carried out on an Illumina MiSeq (250 bp, paired-end) platform in two different batches: a MiSeq V.2 run and a MiSeq nano run. These were processed separately through the first stages of bioinformatics analysis (see below), and then combined after

primer removal for dereplication. PCR replicates (derived from the same sampled bottle of water) sequenced on different runs clustered together without exception (see 'Results'), and thus combining the data from two sequencing runs was appropriate.

## Bioinformatics

We processed the resulting sequence reads with *banzai*, a custom Unix-based script (*O'Donnell, 2015*), which calls third-party programs (*Martin, 2011*; *Zhang et al., 2014*; *Mahé et al., 2015*) to move from raw sequence data to a quality-controlled dataset of counts of sequences from operational taxonomic units (OTUs). A total of 5,105,198 reads survived preliminary quality-control in the bioinformatics pipeline, representing 149,829 OTUs, most of which were rare (<5 reads). We controlled for contamination in three ways, following our approach in *Kelly et al. (2017)*. First, to address the question of whether rare OTUs are a function of low-level contamination or are true reflections of less-common amplicons, we used a site-occupancy model to estimate the probability of OTU occurrence (*Royle & Link, 2006*; *Lahoz-Monfort, Guillera-Arroita & Tingley, 2016*), using multiple PCR replicates of each environmental sample as independent draws from a common binomial distribution. We eliminated from the dataset any OTU with <80% estimated probability of occurrence (a break point in the observed distribution of occupancy probabilities), yielding a dataset of 4,811,014 reads (7,503 OTUs). Second, we estimated (and then minimized) the effect of potential cross-contamination among samples—likely due to tag-jumping (*Schnell, Bohmann & Gilbert, 2015*) or similar effects—as follows: (1) we calculated the maximum proportional representation of each OTU across all control (here, ostrich) samples, considering these to be estimates of the proportional contribution of contamination to each OTU recovered from the field samples; (2) we then subtracted this proportion from the respective OTU in the field samples, yielding 4,370,486 reads (7,496 OTUs). Finally, we dropped samples that had highly dissimilar PCR replicates (Bray–Curtis dissimilarities >0.49, which were outside of the 95% confidence interval given the best-fit model of the observed among-replicate dissimilarities). The result was a dataset of 4,164,517 reads (7,496 OTUs), or 81.57% of the post-pipeline reads. We rarefied read counts from each PCR replicate to allow for comparison across water samples using the vegan package for R (*Oksanen et al., 2015*), such that each sample consisted of $1.85 \times 10^4$ reads from 7,155 OTUs. We carried out subsequent analyses on a single, illustrative rarefaction draw; rarefaction draws did not vary substantially (Fig. S1).

All bioinformatics and other analytical code is included as part of this manuscript, including OTU tables and full annotation data, and these provide the details of parameter settings in the bioinformatics pipeline. In addition, sequence data are deposited and publicly available in GenBank (SRP133847).

## Statistical analysis

### Apportioning variance in Bray–Curtis dissimilarity among sites, sampling events, bottle samples, and PCR replicates

We calculated the variance in OTU communities at five hierarchical levels—between tides (incoming vs. outgoing), among geographic sites ($N = 3$), among sampling events within geographic sites ($N = 4$ per site), among sample bottles within a sampling event ($N = 3$ per

event per site), and among PCR replicates ($N = 3$ per individual sample bottle; reflected by the model residuals)—using a PERMANOVA test on Bray–Curtis (OTU count data) dissimilarity among sequenced replicates. Calculations were carried out in R ver. 3.3.1 (*R Core Team, 2016*) using the vegan (*Oksanen et al., 2015*) package. Having established that the variance among PCR replicates and bottles was small relative to variance among sampling events and geographic sites (see 'Results'), it was clear that our dataset had the necessary resolution to detect community-level changes, if any, associated with changes in tide.

### How many ecological communities are present?

We then used Bray–Curtis dissimilarity to visualize differences among sampled communities at each hierarchical level of organization, using ordination (NMDS, *Venables & Ripley, 2002*), treemaps (*Wickham, 2009*; *Wilkins, 2017*), and a heatmap. Given the strong and consistent differentiation we identified between two ecological communities in the eDNA data (see 'Results'), we then labeled these communities 1 and 2, and applied a set of standard statistics to test for associations between community identity and geographic site (Fisher's exact test), tidal direction (incoming vs. outgoing; $\chi^2$), and tidal height (logistic regression).

Amplification biases inherent in PCR can create amplicon read-counts that differ by orders of magnitude (*Piñol et al., 2015*). Very common taxa (or here, OTUs) disproportionately affect the calculation of Bray–Curtis dissimilarity. Here, we report Bray–Curtis dissimilarities because they capture the kinds of differences researchers are likely to see when using eDNA sequencing as an assay of community similarity. However, presence/absence metrics such as Jaccard similarity, which give proportionately more weight to rare taxa or OTUs, are likely to be more useful in some circumstances expressly because they minimize the effects of amplification bias. We therefore report Jaccard similarities where noted, and include further Jaccard analyses in the supplemental material (Figs. S3 and S5; Table S2).

### Community identity by site and tide

We recovered tidal height data for our study sites during the relevant dates from the National Oceanographic and Atmospheric Administration data for Union, Washington (available at: https://tidesandcurrents.noaa.gov/noaatidepredictions.html).

### Characterizing the observed ecological communities

A single genetic locus provides only a biased and incomplete view of an ecosystem (see *Kelly et al. (2017)* for discussion), and although our purpose was to test for the effect of tidal fluctuations on detected eDNA communities—which does not require taxonomic annotation of the recovered OTUs—we were nevertheless interested in the membership of the ecological communities we detected. Our locus of choice, COI, provided a broad view of ecosystem with 23 phyla in 8 kingdoms represented (see Table S3 for summary table). Algae dominated the read counts, with approximately 91% of annotated reads mapped to taxa in the groups Chlorophyta and Phaeophyceae.

We assigned taxonomy to each OTU sequence using blastn (*Camacho et al., 2009*) on a local version of the full NCBI nucleotide database (current as of August 2017), recovering up to 100 hits per query sequence with at least 80% similarity and maximum *e*-values of $10^{-25}$ (culling limit = 5), and reconciling conflicts among matches using the last common ancestor approach implemented in MEGAN 6.4 (*Huson et al., 2016*). A total of 93.08% of rarefied OTUs could be annotated at some taxonomic level, with over half (57.54%) being annotated to the level of taxonomic Family or lower.

We report an index of community-wide changes across sampling events using the top 10 most-common taxonomic Families in the dataset. We carried out a finer-grained analysis to identify the OTUs driving the observed community shifts at Twanoh by first using a cannonical correspondence analysis (CCA; *Oksanen et al., 2015*), constrained by community identity (1 vs. 2, identified via NMDS; see Results), then filtering the CCA scores by read count, such that we plotted only OTUs that strongly differentiated communities and that occurred at least 1,000 times in the dataset. We then show these by Family-level taxonomic annotation.

## RESULTS

### Apportioning variance in Bray–Curtis dissimilarity

To evaluate the spatial and temporal turnover between eDNA communities, we first apportioned the observed variation in COI Bray–Curtis dissimilarities (calculated using OTU read counts) among tides (incoming vs. outgoing), sampling sites, sampling events within a site, biological replicates (individual bottles of water taken during the same sampling event), and technical replicates (PCR replicates from the same bottle of water). Across the whole dataset, ecological communities at different sampling sites (20–50 km apart) account for the largest fraction of the variance (0.43; Fig. 2), and different sampling events within those sites account for the next highest proportion (0.31). In contrast, biological replicates ($N = 3$ bottles of water per sampling event, taken ca. 10m apart) account for a small fraction (0.07) of the variance, with differences among tides accounting for the smallest fraction of the variance in community dissimilarity (0.06). The remainder—0.13—is largely due to differences among technical PCR replicates ($N = 3$ per bottle of water), much of which derives from stochasticity in the presence of rare OTUs (Fig. S2). The comparatively low variance issuing from biological and technical replicates relative to sampling events and sites affords the resolution necessary to further examine questions of community composition across space and time.

To examine the effect of tide at each of our three geographic locations independently, we again apportioned variance among sampling event, tide, sampling bottles (biological replicates), and PCR replicate (residuals; Fig. 2). Analyzing individual site-level data in this way eliminates the portion of variance due to between-site differences, effectively amplifying the contributions of the remaining hierarchical sampling levels, including tide. Because we treat tidal direction (incoming vs. outgoing) as the highest hierarchical level, we are effectively asking whether eDNA assays reflect a coherent "incoming" tidal community and a coherent "outgoing" tidal community across all sites. For each of our
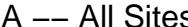

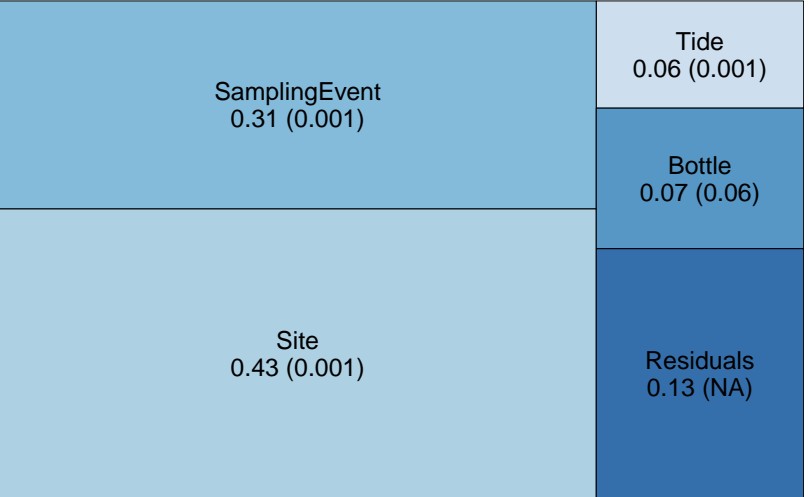

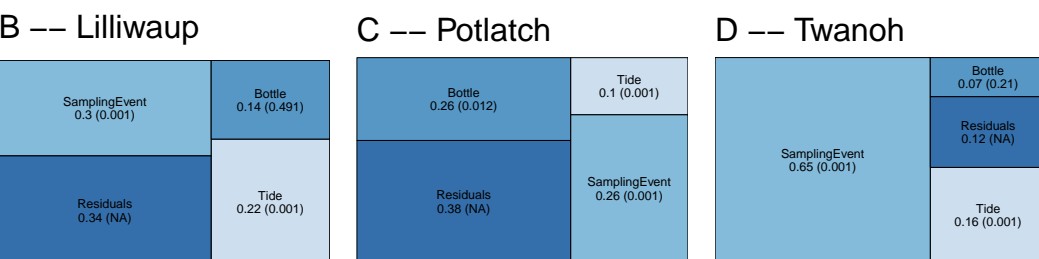

**Figure 2** **Results of PERMANOVA, apportioning variance by hierarchical levels of sampling design: Tide (incoming vs. outgoing), Sampling Site, Sampling Event ($N = 4$ time points per site), and Sampling Bottle ($N = 3$ bottles per sampling event).** Residuals reflect variance among PCR replicates ($N = 3$ replicates per sampling bottle) as well as variation due to rarefaction stochasticity and other sampling effects. (A) reflects results for the dataset as a whole, with (B–D) giving site-specific variances. Numbers reflect proportion of the variance explained by the indicated hierarchical level ($R^2$), with permutation-derived $p$-values in parentheses.

three sampling locations, variation among sampling events remains greater than variation between incoming and outgoing tides (Fig. 2B), with little evidence of consistent incoming or outgoing tidal communities. Using Jaccard distances (OTU presence/absence) rather than Bray–Curtis similarly apportions the smallest fraction of variance to tidal direction (0.02, $p = 0.001$; see Supplementary Information for further Jaccard analyses).

We next tested the possibility that eDNA sequences might still regularly shift in association with tide, even if not between two predicable assemblages (see above). Conceiving of tidal turnovers within a site as a series of events that could each influence community composition, we treated our first sampling event at a site as the reference point for that site, and assessed the Bray–Curtis dissimilarity of eDNA sequences with each subsequent sampling event occurring at a later point in time and after one or more changes

in tide (Fig. 3; this technique is known as pseudo-autocorrelation; see *Fuhrman, Cram & Needham, 2015*). If ecological communities within each site remain consistent over time, we expect the Bray–Curtis values of the community at time zero (the reference community) vs. time one (the subsequent sampling event) to be identical to the dissimilarity values among bottles taken within the same sampling event. We observe little change in community dissimilarity as a function of tidal change (or indeed, of time).

In all three sites, Bray–Curtis values remain stable across multiple tide changes, with no continuously increasing trend over time. Instead, two events stand out as statistically significant (Kruskal–Wallis, $p < 0.01$): a moderate increase at Lilliwaup at time step 3–ca. 26 h after the reference sample—from median 0.26 to 1), and a far larger jump in a single time point at Twanoh (ca. 19 h after reference; from 0.2 to 0.72, before returning to its reference value in the subsequent sampling event). In each of these events, a change in salinity of the sampled water is significantly associated with the change in ecological community, while time-since-reference is not (linear models; Lilliwaup $t$-value Salinity = 3.96 and Time-since-reference = 0.65; Twanoh $t$-value Salinity = 3.63 and Time-since-reference = 0.17; note that time-since-reference necessarily encompasses tidal changes in our sampling scheme; See Fig. S4 for site-level regressions with between Bray–Curtis dissimilarities and changes in salinity). See Fig. S5 for similar results using Jaccard distances rather than Bray–Curtis.

In sum, neither tidal direction (incoming vs. outgoing) nor individual tidal events therefore consistently drives differences in sampled eDNA communities, but as described below, changes in water masses such as those seen in the Twanoh changeover event bear further scrutiny.

## How many ecological communities are present?

We created an ordination plot of Bray–Curtis distances among each of our sequenced replicates to visualize any distinct ecological communities present in the dataset (Fig. 4A). In agreement with the analysis of variance, technical PCR replicates and biological replicates consistently cluster closely in ordination space, yet two non-overlapping eDNA sequence assemblages appear on this plot. A heatmap of the same Bray–Curtis values reveals the underlying magnitudes of dissimilarity and clustering, showing two clearly distinct communities of eDNA sequences (Fig. 4B). The two observed clusters are primarily associated with sampling site: the left-hand community (ordination plot; Fig. 4A) is present in all technical and environmental replicates of all Lilliwaup and Potlatch samples, and in all such replicates from a single Twanoh sampling event. We call this "community 1" below. By contrast, the right-hand community ("community 2") is only present in the remaining three Twanoh samples. Jaccard-based NMDS analyses show very similar patterns, including identifying the two distinct communities (Fig. S3).

## Community identity by site and tide

To further investigate the relationship of each eDNA community with tide, we first assigned membership of each sample to one of the two communities identified in our ordination analysis (Fig. 4A) and plotted community membership of each sample across the tidal

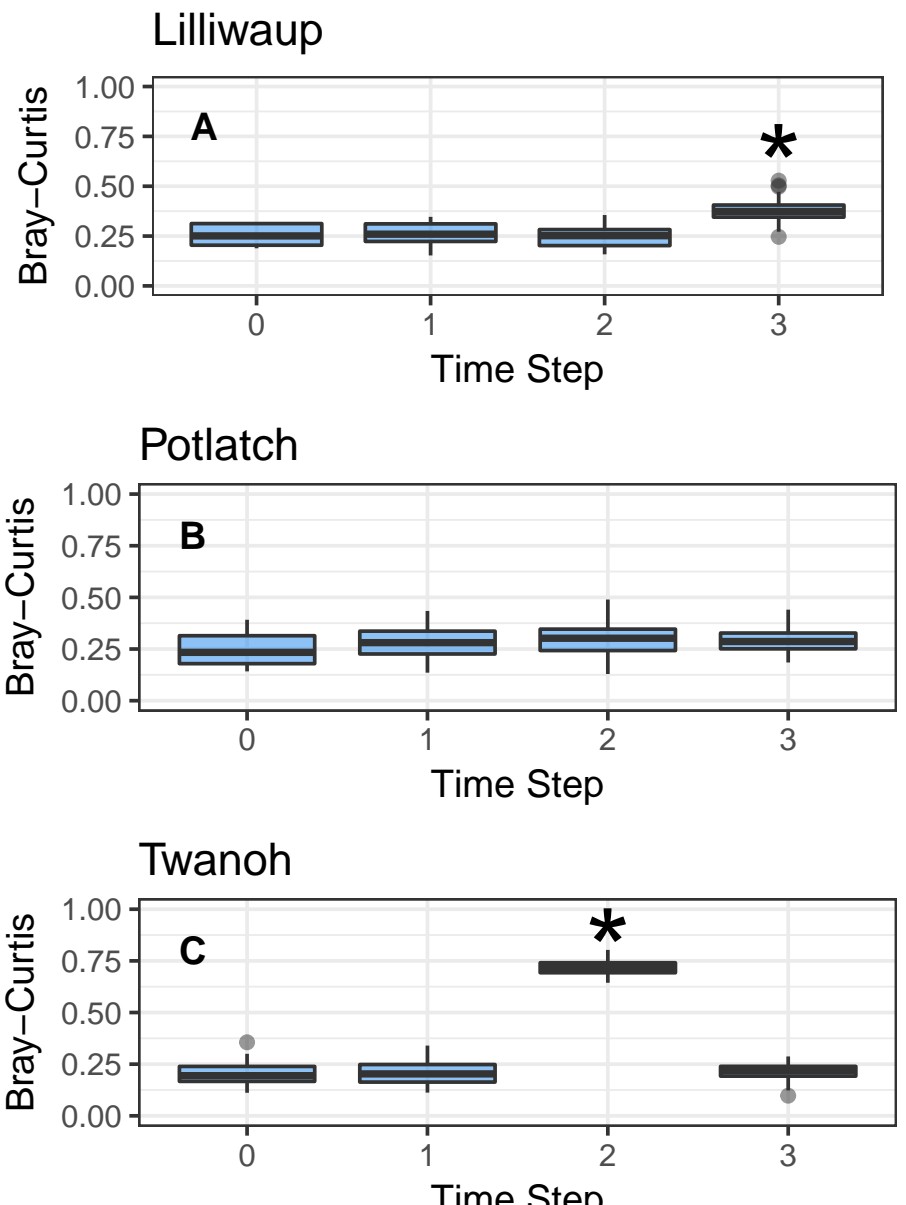

**Figure 3** Comparison of Bray–Curtis dissimilarities within a reference sampling event (Time step = 0) and between the reference sample and subsequent samples at the same site (Time Steps 1, 2, and 3). Subsequent time steps reflect the accumulation of ecological eDNA differences over hours as the tide moves in and out. Sites shown individually. Steps with significant increases (Kruskal; $p < 0.01$) marked with asterisks and discussed in the text. $Y$-axes identical to facilitate comparison across sites. (A–C) show sites Lilliwaup, Potlatch, and Twanoh, respectively.

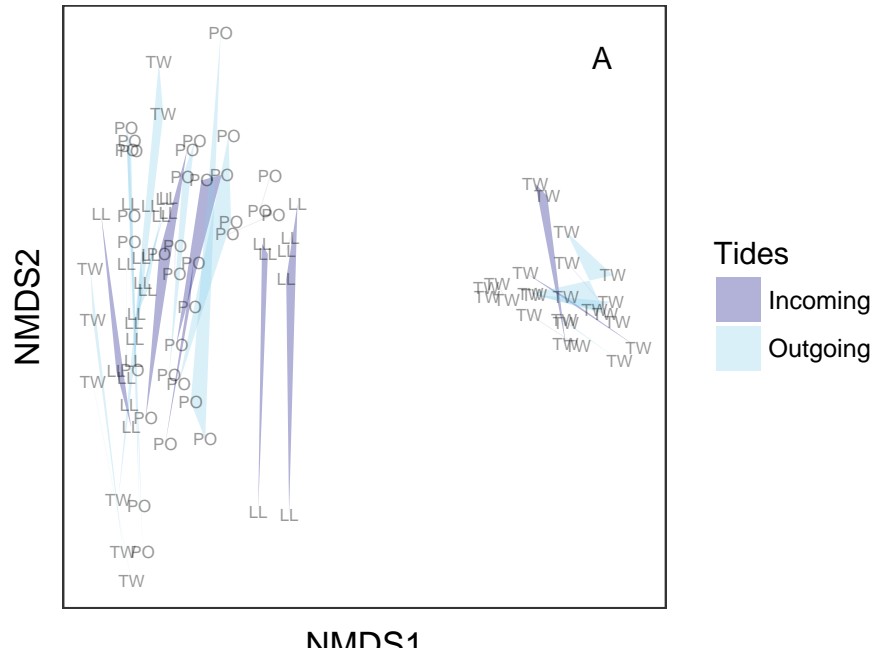

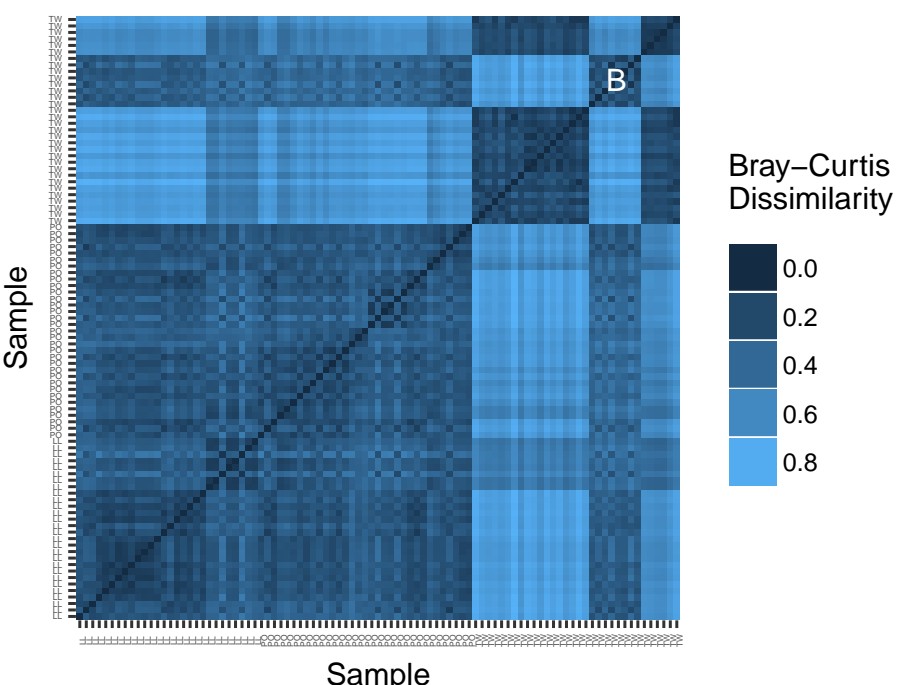

**Figure 4** **(A) Ordination plot (non-metric multidimensional scaling; NMDS) plot of Bray–Curtis dissimilarities among sequenced replicates, by sampling bottle (polygon) and tide (polygon color). Polygons connect communities sequenced from replicate PCR reactions of the same sampled bottle of water. (B) The same data shown as a heatmap, ordered by site identity. Only the Twanoh samples (upper right) stand out as having substantial heterogeneity, reflecting the two different communities present during different sampling events at that site.** Site labels: TW, Twanoh; PO, Potlatch; LL, Lilliwaup.



 

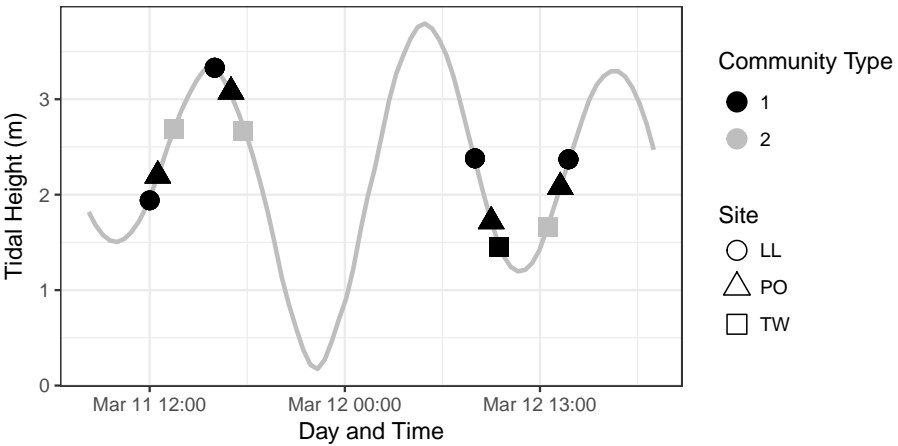

**Figure 5** **eDNA Communities by Time, Tide, and Site.** We identify community type 1 as the dominant eDNA community (as seen in Fig. 4, which appears at every geographic site), and community type 2 as the distinct type occurring only at Twanoh. See text for community descriptions.

cycle during collection (Fig. 5). Both figures qualitatively indicate a lack of association between tidal direction or height and either of the two eDNA communities. Quantitatively, by sampling event, community is independent of tidal height ($p = 0.39$; linear regression) and of tidal direction (incoming vs. outgoing; $p = 0.163$; $\chi^2$), but is related imperfectly to site identity ($p = 1.554e{-}15$; Fisher's exact test). The fact that Twanoh hosts different communities at different times indicates that geography does not fully explain differences between these communities, and that ecological variables warrant further investigation as driving differences in communities.

## Environmental covariates assocated with community change

To identify ecological factors that might distinguish the two eDNA assemblages observed, we modeled the association of each sample's temperature, salinity, and site identity with communities 1 and 2. Salinity and temperature explain nearly all of the variance in community type (logistic regression best-fit model; null deviance = 84.79, residual deviance = 1.033e–09): we observe community 2 in fresher (<20 ppt salinity) and colder (<9 °C) water than we find community 1. Twanoh, in the southeastern portion of Hood Canal most distant from the ocean, routinely experiences these kinds of fresher, colder water events in our sampling month (March), unlike the main stem of the Canal (Fig. S3).

In summary, the eDNA communities are more closely associated with water mass—or perhaps with water-mass-associated ecological variables such as salinity and temperature—than with tide, or even with geographical origin. This observation led us to investigate the taxonomic composition of our identified communities. We summarize the ecological and biological context of each community sample in Fig. 6, before highlighting the taxa that are particularly influential in defining the two communities. In addition, to demonstrate that our observed results are not simply a function of trends in single-celled planktonic taxa (expected to change with water mass), we provide Family-level analyses in Figs. S7 and S8 showing that (1) tide accounts for the smallest fraction of eDNA variance in nearly

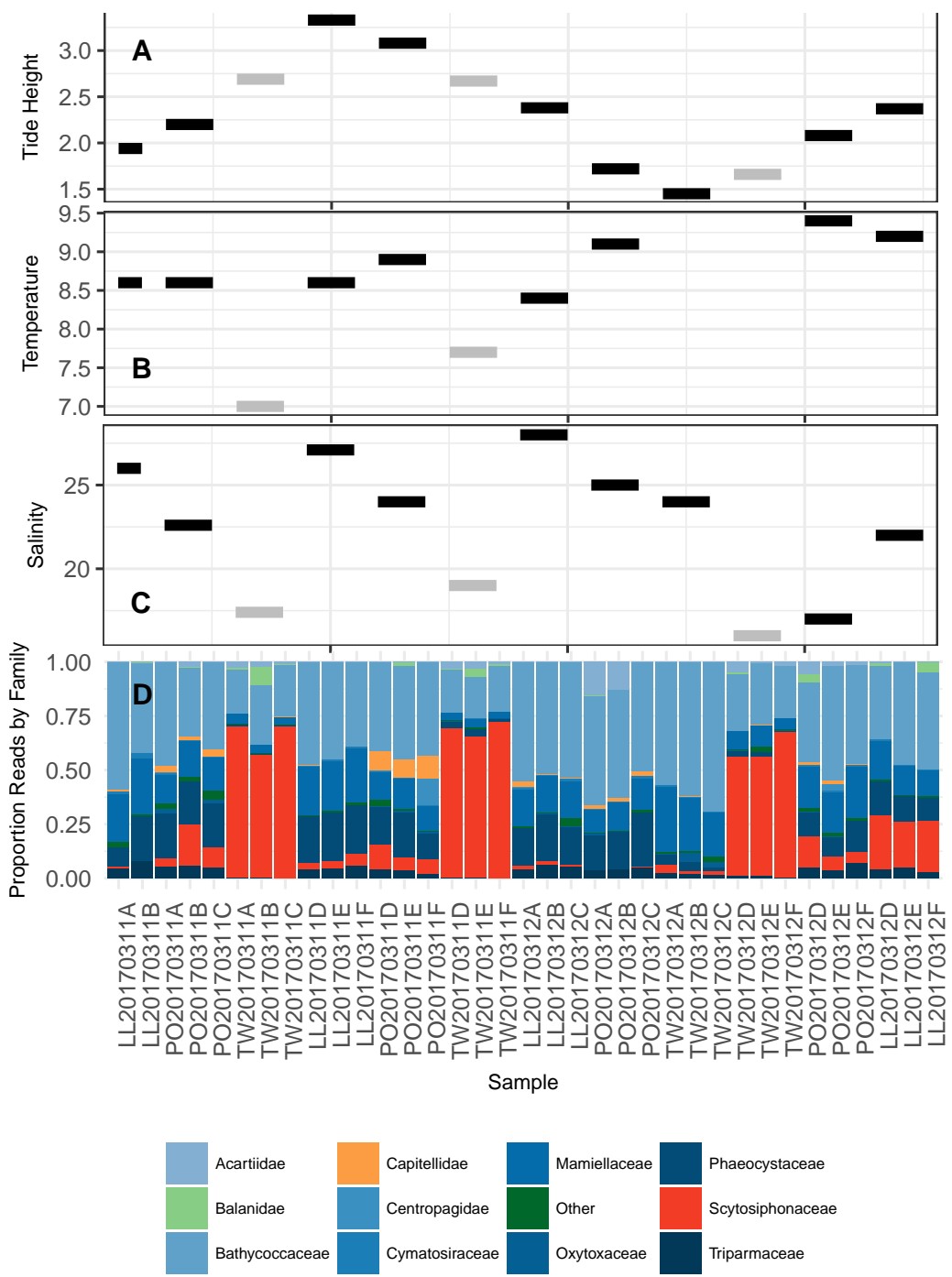

**Figure 6** **Tidal height (A; meters), Water temperature (B; degrees C), Salinity (C; ppt), and proportion of DNA reads allocated among the most common Families in the annotated dataset (D).** Color coding in subplots (A–C) reflect the community types in Fig. 5 (Black = 1, Grey = 2). Two temperature points are missing in subplot (B) due to a failure of equipment. Color coding in subplot (D) is as follows: primarily planktonic taxa in shades of blue, primarily non-planktonic taxa in shades of orange, taxon with prominent planktonic and non-planktonic stages (Balanidae) in light green, and "other" in dark green.

every Family, and (2) eDNA associations with water mass (here, indexed by salinity) occur across Families and phyla with both benthic and planktonic life histories.

## Taxa associated with distinct communities

To identify the taxonomic groups that most strongly differentiate ecological communities 1 and 2 at Twanoh, the location at which we detected both communities at different points in time, we performed a constrained canonical correspondence analysis (CCA) principal component analysis on the OTU counts. We constrained the ordination by community identity as determined by the distance analyses above and filtered for highly discriminating OTUs with high read counts (>1,000 reads) to identify a set of high-leverage taxa distinguishing communities. The result was seven Families (Fig. 7), two of which are animals—Balanidae (barnacles; benthic as adults, planktonic as larvae) and Acartiidae (copepods; planktonic)—and the others of which are autotrophic groups consisting of dinoflagellates (Oxytoxaceae; planktonic), chlorophytes (Mamiellaceae, Bathycoccaceae; planktonic), a sessile brown alga (Scytosiphonaceae; benthic), and a another heterokont, Triparmaceae (planktonic). A handful of benthic and planktonic taxa therefore distinguishes the two communities we identify with COI. However, given the well-known effects of primer bias—by which the apparent abundance of some taxa can be grossly distorted by equating read count to organismal abundance—we stress that here we are using a single primer set as an index of community similarity, rather than as an accurate reflection of the abundances of taxa present in the water.

## DISCUSSION

Environmental DNA is rapidly becoming an essential and widely-used tool to identify community membership in aquatic environments (*Taberlet et al., 2012*; *Spear et al., 2015*; *Thomsen & Willerslev, 2015*; *Kelly et al., 2016*; *Yamamoto et al., 2016*; *Deiner et al., 2017*). But it is not yet clear to what extent the sequences identified in eDNA studies reflect the presence of local organisms in time and space (*Jane et al., 2015*; *Port et al., 2016*; *Wilcox et al., 2016*). Of particular interest in marine systems is the influence of tide on the detection of ecological communities: must sampling schemes standardize tidal height and direction during collection to detect consistent groups of species? Does each tide bring with it a turnover in water, carrying exogenous DNA, or do the sequences detected at any given time accurately reflect the species present within a habitat in that moment? But more generally for eDNA studies, to what extent must we worry about where DNA comes from and where it goes? To address these questions, we collected and analyzed eDNA communities at three different sites along the Hood Canal over the course of multiple tidal turnovers. Thus, for each site, we were able to examine the influence of reversals in tidal direction and larger-scale changes in the water present at our study sites.

When analyzed together, eDNA collections from three locations (Lilliwaup, Potlatch, and Twanoh) show substantial variance in OTU membership and prevalence associated primarily with geographic location (Fig. 2). Grouping of samples in ordination space is also strongly associated with site, rather than with tide (Fig. 4A). Together, these results suggest that eDNA surveys designed to clarify relationships between distinct ecological

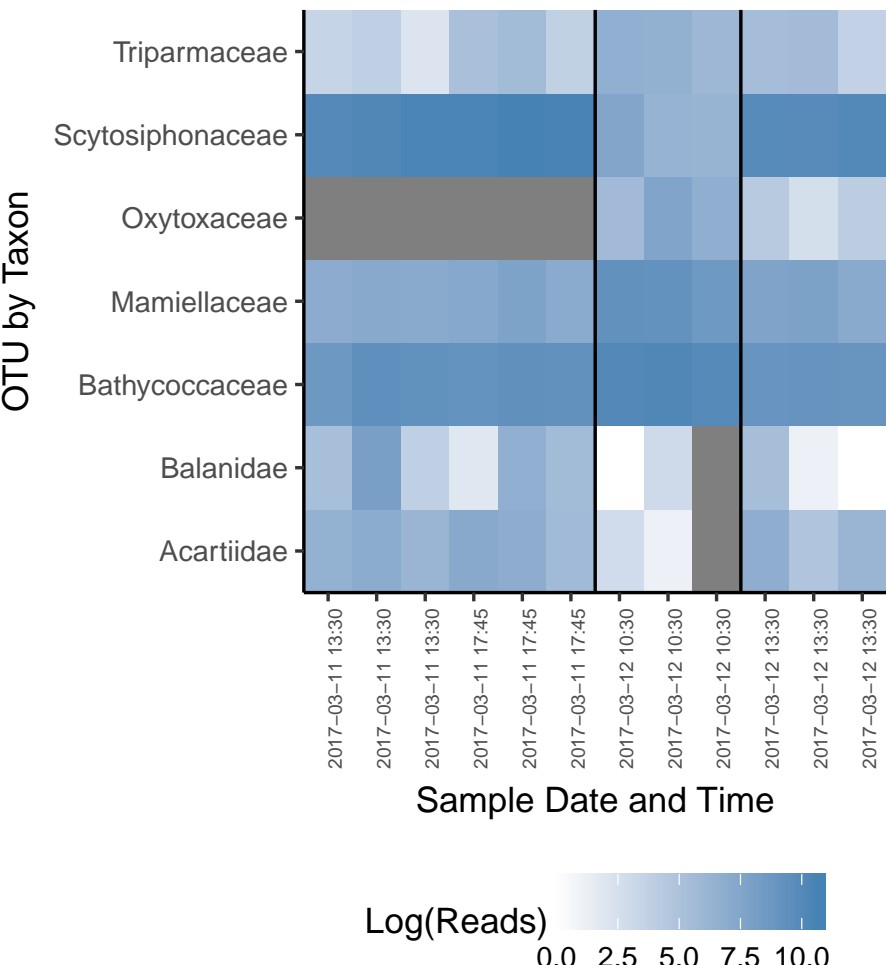

**Figure 7** **Most influential OTUs, plotted by taxonomic Family, distinguishing the two ecological communities observed in water samples from Twanoh State Park.** Shown are the taxonomic Families of OTUs with at least 1,000 reads in the rarefied dataset, and having a constrained canonical correspondence analysis (CCA) score of greater than 0.7 (absolute value), which in our dataset most clearly divides the two communities. See text for CCA details. The vertical black lines in the chart delineate the communities identified previously by NMDS (see Fig. 3A), with the time of each sample given along the *x*-axis, showing the shift from one community to the other—and then largely back again—within less than 24 h. Note that each block of samples reflects a different point in the tidal cycle, and that the first two time points indicate a continuity of community membership despite a change in tide (see Fig. 4).

communities are not likely to suffer substantially from sample collection at varying points in the tidal cycle, because the twice-daily exchange of water into- and out of our sampling sites appeared to have little influence on the sequences detected overall.

Although the effect of tide on eDNA community composition is small when multiple geographic sites are considered simultaneously, tidal direction may still influence the OTUs detected within a single location. The existence of among-site differences in

ecological communities in fact provides the resolution necessary to detect such a local influence of tide, if present—exogenous DNA arriving periodically with tidal flow at each site might closely resemble neighboring communities, and differ consistently from endogenous DNA collected on the ebb tide, which has spent hours in contact with local benthic flora and fauna. A site-by-site analysis reveals that a significant proportion of the variance in OTU counts is associated with tide, but never as much as is associated with differences between sampling events (Fig. 2). These results suggest community variance among individual sampling events, although small in an absolute sense, dominates changes at the site scale and accordingly that there is no coherent incoming- or outgoing-tide eDNA fauna. Additionally, the eDNA community present at a single site tends to drift little over time and with successive tidal turnovers (Fig. 3), instead changing in association with changes in salinity and temperature of the water mass present at the time of sampling (Fig. 6, Fig. S3). Together, these results suggest that the effect of tidal flow, *per se*, on eDNA community membership is minimal relative to the differences associated with changes in water characteristics and geographic site.

Rather than tide, ecological variables such as temperature and salinity, each of which differ among sites and sampling events, drive the bulk of the variance in eDNA community membership (Fig. 6 and multiple regression). At Twanoh, we sampled by chance a dramatic shift in species composition from community 2 to community 1 within the span of just a few hours, and a concomitant shift towards warmer, more saline water relative to baseline. Of the seven families most notably associated with this turnover, four single-celled planktonic taxa (Triparmaceae, Oxytoxaceae, Mamiellaceae, and Bathycoccoceae) increased in OTU count with intrusion of the warmer, saltier water mass. By contrast, two families with benthic adults (Balanidae and Scytosiphonaceae) and one planktonic animal (Arctiideae) decreased (Fig. 7). More generally, we saw Family-level associations with salinity across a wide variety of taxa and life-histories (Fig. S8). These results broadly suggest that this particular eDNA survey methodology succeeds in identifying changes in the planktonic species physically present within the water column at the time of sampling as well as detecting benthic or sessile species; the entrance and exit of a water mass with characteristics more common at neighboring sites diminishes (but does not eradicate) the signal from non-planktonic groups. The sequenced eDNA community therefore reflects contributions from both organisms living within the water itself, as well as immobile species in contact with that more mobile community.

## CONCLUSION

Taken together, our results suggest that eDNA samples from even highly dynamic environments reflect recent contributions from local species. With the exception of the occasional movement of water masses representing distinct habitats for planktonic organisms, the eDNA communities we sampled at three geographic sites were largely stable over time and tide. Practically, this suggests that intertidal eDNA research should be performed with substantial attention to ecological variables such as temperature and salinity, which serve as markers of the aqueous habitat present and which may not remain

consistent geographically. In contrast, tidal turnover itself appears to be a secondary consideration that does not dramatically or consistently affect the community sampled, even within a single geographic location. Marine intertidal eDNA surveys therefore appear to reflect the endogenous DNA of the organisms present in the water and on the benthic substrate at the time of sampling.

## ACKNOWLEDGEMENTS

We thank K Cribari for lab assistance, as well as R Morris, G Rocap, and V Armbrust at the UW Center for Environmental Genomics. Special thanks to M Kelly for access to the field station, A Ramón-Laca and E Flynn for facilitating fieldwork, and Julieta, Damián, and Owen for field assistance. We are also grateful to L Park, J O'Donnell, K Nichols, and P Schwenke at NMFS for sequencing support and expertise. We thank the reviewers for improving the piece substantially.

### Funding

This work was made possible by grant 2016-65101 from the David and Lucile Packard Foundation to Ryan Kelly. The funders had no role in study design, data collection and analysis, decision to publish, or preparation of the manuscript.

### Grant Disclosures

The following grant information was disclosed by the authors:
David and Lucile Packard Foundation: 2016-65101.

### Competing Interests

The authors declare there are no competing interests.

### Author Contributions

- Ryan P. Kelly performed the experiments, analyzed the data, contributed reagents/materials/analysis tools, prepared figures and/or tables, authored or reviewed drafts of the paper, approved the final draft.
- Ramón Gallego conceived and designed the experiments, performed the experiments, analyzed the data, contributed reagents/materials/analysis tools, prepared figures and/or tables, authored or reviewed drafts of the paper, approved the final draft.
- Emily Jacobs-Palmer analyzed the data, contributed reagents/materials/analysis tools, prepared figures and/or tables, authored or reviewed drafts of the paper, approved the final draft.

### Data Availability

The raw data are available on Genbank (SRP133847), with relevant metadata and code also available at https://github.com/invertdna/eDNA_Tides.

## Supplemental Information

Supplemental information for this article can be found online at http://dx.doi.org/10.7717/peerj.4521#supplemental-information.

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
