# Peer review of "The effect of tides on nearshore environmental DNA"

_PeerJ, doi:10.7717/peerj.4521_

## Round 0.1 · original submission · Major Revisions

We have now received three reviews. Although the three reviewers agree that your work is interesting and worth of publication, they differ substantially in the extent of the changes necessary for it to be acceptable for publication.

Reviewers #1 and #2 were more enthusiastic about your work than Reviewer #3. After careful reading of the reviews and of the manuscript itself, I agree with his/her (and also partially Reviewer #1) major concern about the origin of the DNA. I consider he/she makes a very important point about the representation of pelagic (mostly planktonic protists) vs. benthic organisms (metazoans and macrophytes) in your eDNA. I urge you to pay special attention to this comment and accordingly address the balance between planktonic and benthic eukaryotes, involving a more detailed scrutiny of species composition as suggested by Reviewers #1 and #3. This will help better focus your study. Reviewer #2 highlights his/her concern about the negative filter controls and the possible PCR bias suggesting a more conservative approach of presence/absence rather than abundance for assessing differences between communities.

Additionally, all reviewers have identified several technical and editorial issues that need to be carefully considered in the revision. Please prepare a revised version of your manuscript detailing how you have addressed each comment by the reviewers on a point-per-point basis. Depending on the way you have addressed Reviewer #3’s comments I may have him/her look at it again.

We are looking forward to receiving your revised manuscript.

Reviewer 1 ·

Basic reporting

No comment.

Experimental design

No comment.

Validity of the findings

No comment

Additional comments

General comments
The need for the research is clearly stated, the study design is clear, the analyses are rigorous and easy to follow, and the conclusions do not extrapolate beyond the results. Taken in total, this is an outstanding contribution that I have very little to be critical of. I strongly advocate for the manuscript’s publication in PeerJ. I have only two discussion points I would like the authors to consider.

At each sampling location (Lilliwaup, Potlatch, and Twanoh) three samples were collected approximately 10 m apart. My experience with intertidal areas, along with my expectation that there is considerable water mixing at the nearshore would lead me to a priori expect that water samples taken this close together would necessarily be very similar, both on the incoming tide and outgoing tide. Finding consistency among samples at a site could be a function simply of a well-mixed system and not necessary a reflection of local species. Alternatively, do deep water species (likely not in the intertidal area) showing up in the samples? I suggest more scrutiny of the species detected and their likely location in the system is justified to provide stronger evidence for the conclusions. I fully agree that between the three locations there are differences in species composition, and the authors have solid ground to talk about localized scale of inferences.

Additionally, Shogren et al. (2017) talked about retention of DNA. What role would retention of eDNA trapped in tidepools, on macrophytes, trapped in the sand and then re-suspended play in the detection signal?

Shogren, A.J., Tank, J.L., Andruszkiewicz, E., Olds, B., Mahon, A.R., Jerde, C.L. and Bolster, D., 2017. Controls on eDNA movement in streams: Transport, retention, and resuspension. Scientific Reports, 7(1), p.5065.

I don’t know the answer to this, but given the potential wave action influence going could the low tide shift to hi tide purely be a legacy effect of DNA being “trapped?”

·

Basic reporting

This manuscript is clearly written and the literature citations are very thorough. The article structure, figures, and tables are appropriate and clear, and the raw data will be shared upon publication. The results are consistent with the hypothesis the nearshore eDNA communities change in unpredictable ways.

Experimental design

This is an interesting ecological/environmental study that is appropriate for PeerJ. The research question ("Do eDNA communities change with tides in nearshore coastal environments") is well defined throughout the manuscript and fills in a significant gap in our knowledge of eDNA dynamics. The experimental design was appropriate and, although a few more sites would have been ideal (3 sites included in the study), sufficient to answer the main question. The eDNA collection, amplification, sequencing, bioinformatics, and statistical protocols all followed best practices in the field.

My only methodological concern is with the treatment of negative filter controls. The statement of Page 3, Line 91 "... but did not sequence them [negative controls] due to the practical and theoretical issues associated with library preparation in samples without any discernable amplicon." needs a citation. I know of many studies that do sequence these samples, even if no PCR band is present. Also, given the high possibility of cross-contamination at the sample collection or filtering stages of this study, skipping the sequencing of the negative filter controls is an issue. Also, please note somewhere that using the Ostrich samples to gauge cross-contamination won't detect contamination that occurred during sample collection or filtering stages, just lab contamination.

Validity of the findings

no comment

Additional comments

Overall this was a well-written and rigorous study that provides new insight into the "ecology" of nearshore eDNA communities. I have only two few minor comments/suggestions in addition to the negative filter controls concern mentioned previously.

1) Given that PCR bias is inherent in any metabarcoding study (which the authors do a nice job of explaining), shouldn't a presence/absence dissimilarity metric (e.g. Jaccard) be used instead of the abundance-based Bray-Curtis? As the authors say (Page 10, Line 255), the read count abundances don't reflect the true abundances of the organisms.

2) Page 11, Line 277: change "provide" to "provides"

Reviewer 3 ·

Basic reporting

This manuscript uses molecular surveys to compare the biodiversity of eukaryotes living in nearby coastal sites, with the emphasis to see if communities are changing during different times of the tidal cycle. The authors identify two different communities, that are characteristic of different sites. The paper is well written and well presented, but I have a major concern relating with the focus of the study.

My main concern is the poor definition and understanding of what environmental DNA (eDNA). Reading the introduction, the authors seem to imply mostly dissolved or extracellular DNA, that leaves remnants of living cells. The comes from the use of references mostly from metazoans, and specially from the first sentence in the introduction: "organisms leave genetic traces". But then, the data derives from planktonic eukaryotes that live in 1 liter of water and are larger than 0.45 µm. So, basically the thousands of planktonic protists in this water mass (and the few animals that by chance may be collected in this liter of water). So, this is essentially a study on planktonic protists. And on top of that, there is the signal of dissolved DNA released as genetic traces from benthic fauna and flora. The first and needed analysis here would be to separate, as much as possible and based on the taxonomic identification of the sequences, the signal from planktonic microbes and from benthic organisms.

This is very important for the conclusions, as the molecular signal has two components, one that is static and depends on the benthic organisms, and another that is dynamic and depends on the microbes that the water mass is carrying (always they come and they leave the site).

Other comments:

Line 169. "etc" in a heading does not sound right

Line 201. The range "0.26 to 1" is not right

Line 209. Instead of "individual environmental changes" what seems to happen in this individual point is "a water mass replacement", bringing the community that lives in the other site. The same argument in line 238

Line 231. Typo in Associated

Line 262-264. Again this sentence represents the main problem of the manuscript: what is the habitat here? Studying planktonic eukaryotes, the habitat is moving with the tides, so the habitat is itself "exogenous". This seems to arrive at the end of the manuscript, as a surprise (lines 296-297), but in fact is a well known fact that collecting planktonic microbes on filters is the standard way to study their biodiversity.å

Fig 6, bottom. I suggest here to add an extra category with the signal of the "other groups". This will bring a more fair comparison of the different families

Experimental design

The design is fine and appropriate for the actual study

Validity of the findings

As said before, the main problem here is to differentiate between the signal coming from planktonic microbes and benthic organisms.

---

## Round 0.2 · Major Revisions

As you will see in the attached report by Reviewer#3, s/he feels you have not adequately addressed her/his major concern of the distinction about planktonic and benthic organisms, and their relative contribution in your eDNA samples. I must say that your current treatment of this issue and the implications for data analysis and interpretation in the revised manuscript is still not satisfactory. Therefore, I would urge you again to pay full attention to both reports by the reviewer and submit a substantially changed manuscript. I envision this will result not only as changes in the text but also in the presentation of results. I may request the opinion of another reviewer before making a final decision.

Reviewer 3 ·

Basic reporting

xx

Experimental design

xx

Validity of the findings

xx

Additional comments

My main concern of this paper, an important one, was that the authors neglected what I think could be the most important source of their molecular signal. To me, the molecular analysis of what is retained after filtering 1 liter of seawater through a 0.45 µm filter targets mostly microbes that lived suspended in seawater (millions of cells). And this has strong implications to the results shown here. Certainly, there can be signal from other sources (dissolved or detrital DNA retained in the filter, small animals that lived in this particular liter of water, or fragments of animals and macroalgae). But I still think that this "organismal" DNA is more important than the "genetic traces" of benthic organisms. And at any rate, the relative importance of both elements needs to be stated.

In their response, the authors seem to be aware of the existence of planktonic microbes and make a long answer to this general comment, but then when I look at the manuscript, the paper looks essentially the same. So, this essential fact has not been mentioned at the introduction and is very superficially treated in the general conclusions.

Just in the abstract there are good examples of how my main concern has not been addressed:
"Organisms of all kinds leave behind genetic information in their environments"
Yes, there is the genetic signal left by organisms but, on top of that and likely more importantly, there is the signal of the organisms (in this case microbes) living in the actual sample taken.
"Our findings suggest that nearshore eDNA tends to be endogenous to the site and water mass sampled, rather than changing with each tidal cycle"
Given that the sample analyzed has two components (and this should be explicitly mentioned, treated, presented, and discussed), I disagree with this sentence. There is a component that lives in the water, so it comes and goes with the tides. If the site is the "geographic coordinate", this component is exogenous by definition. And then there is the component that derives from the benthic organisms, which are less mobile and therefore essentially endogenous.

---

## Round 0.3 · accepted · Accept

I appreciate the efforts you have made to address the last concerns of Reviewer #3 and myself. I have just one final comment that I hope you will be able to correct before the final version of your article is published. Please eliminate the column "Kingdom" in Supplemental Table 3 since there is no consensus about this level of classification and phyla already provide all the relevant information. I decided to ask you this here rather than returning it for minor revision that it would unnecessarily delay its publication.